# Predictors of Mortality in Tocilizumab-Treated Severe COVID-19

**DOI:** 10.3390/vaccines10060978

**Published:** 2022-06-20

**Authors:** Konstantinos Pagkratis, Serafeim Chrysikos, Emmanouil Antonakis, Aggeliki Pandi, Chrysavgi Nikolaou Kosti, Eleftherios Markatis, Georgios Hillas, Antonia Digalaki, Sofia Koukidou, Eleftheria Chaini, Andreas Afthinos, Katerina Dimakou, Ilias C. Papanikolaou

**Affiliations:** 1Pulmonary Department, Corfu General Hospital, 49100 Corfu, Greece; kpag28@otenet.gr (K.P.); manos24m@yahoo.gr (E.A.); kellypandi@gmail.com (A.P.); lefte_mark83@yahoo.gr (E.M.); elhaini@otenet.gr (E.C.); andreasafthinos@gmail.com (A.A.); 25th Respiratory Medicine Department, SOTIRIA Chest Hospital, 11527 Athens, Greece; makischr@hotmail.com (S.C.); xkdocxk@gmail.com (C.N.K.); ghillas70@yahoo.gr (G.H.); tonia.dgl@gmail.com (A.D.); sofiek90@hotmail.com (S.K.); kdimakou@yahoo.com (K.D.)

**Keywords:** COVID-19, tocilizumab, mortality, outcomes, severity

## Abstract

Purpose: Tocilizumab is associated with positive outcomes in severe COVID-19. We wanted to describe the characteristics of nonresponders to treatment. Methods: This was a retrospective multicenter study in two respiratory departments investigating adverse outcomes at 90 days from diagnosis in subjects treated with tocilizumab (8 mg/kg intravenously single dose) for severe progressive COVID-19. Results: Of 121 subjects, 62% were males, and 9% were fully vaccinated. Ninety-six (79.4%) survived, and 25 died (20.6%). Compared to survivors (S), nonsurvivors (NS) were older (median 57 versus 75 years of age), had more comorbidities (Charlson comorbidity index 2 versus 5) and had higher rates of intubation/mechanical ventilation (*p* < 0.05). On admission, NS had a lower PO_2_/FiO_2_ ratio, higher blood ferritin, and higher troponin, and on clinical progression (day of tocilizumab treatment), NS had a lower PO_2_/FiO_2_ ratio, decreased lymphocytes, increased neutrophil to lymphocyte ratio, increased ferritin and lactate dehydrogenase (LDH), disease located centrally on computed tomography scan, and increased late c-reactive protein. Cox proportional hazards regression analysis identified age and LDH on deterioration as predictors of death; admission PO_2_/FiO_2_ ratio and LDH as predictors of intubation; PO_2_/FiO_2_ ratios, LDH, and central lung disease on radiology as predictors of noninvasive ventilation (NIV) (a < 0.05). The log-rank test of mortality yielded the same results (*p* < 0.001). ROC analysis of the above predictors in a separate validation cohort yielded significant results. Conclusions: Older age and high serum LDH levels are predictors of mortality in tocilizumab-treated severe COVID-19 patients. Hypoxia levels, LDH, and central pulmonary involvement radiologically are associated with intubation and NIV.

## 1. Introduction

As of 31 January 2022, a novel coronavirus designated severe acute respiratory syndrome coronavirus-2 (SARC-CoV-2), which causes COVID-19, is responsible for 5,658,702 deaths worldwide. Infection fatality rates vary by geographic location, age, risk factors, and vaccination status, ranging from 0.15% to 1% in unvaccinated individuals but exceeding 15% in humans >85 years of age [1,2].

In-hospital fatality rates also vary, as a mortality rate of 11.4% was reported in a United States 2021 survey, while a mortality rate of approximately 30% was anticipated in critical COVID-19 cases [3,4]. Proinflammatory cytokines such as IL-6 are implicated in severe COVID-19. Tocilizumab, an anti-IL-6 receptor antagonist used for the treatment of rheumatoid arthritis, has shown efficacy against COVID-19 in two large randomized trials. The RECOVERY trial showed a reduction in the mortality rate from 35% to 31% in conjunction with corticosteroids in hospitalized patients with moderate, severe, and critical COVID-19 with evidence of inflammation [5]. In the REMAP-CAP trial, critical COVID-19 cases treated with tocilizumab or sarilumab had a mortality rate of 27% versus 36% in those who received only usual care [6]. Three subsequent meta-analyses supported the use of tocilizumab against COVID-19, the first reporting a pooled mortality prevalence of 19% in the tocilizumab arm, the second a reduced 28-day mortality (RR 0.89, 95% confidence interval 0.82 to 0.97), and the latter a 28-day mortality risk of 22% (Odds Ratios 0.83, 95% CI, 0.74–0.92) [7,8,9].

On the other hand, tocilizumab is clearly not effective in all COVID-19 cases. It has been postulated that early drug administration may be beneficial [10]. In Greece, health authorities recommend its use in worsening severe or critical COVID-19 with evidence of inflammation (with severe COVID-19 defined by the World Health Organization as resting oxygen saturation <90% on room air or <94% plus respiratory rate >30/min and/or lung infiltrates >50% of total lung fields) [11]. Tocilizumab is a potent anti-inflammatory drug with this effect found by others to reflect in lowering c-reactive protein (CRP), albeit not always with clinical benefit, while additional clinical parameters such as the level of hypoxia may be of importance to determine treatment escalation or to determine prognosis, as suggested by other authors [12,13]. The purpose of the present study is to elaborate on the characteristics of patients who received the drug, on clinical indices associated with responsiveness or non-responsiveness to tocilizumab and identify predictors of mortality and of other adverse outcomes related to its use.

## 2. Materials and Methods

### 2.1. Study Design

This is a multicenter retrospective study of COVID-19 subjects hospitalized from May 2021 until November 2021 in two dedicated respiratory departments at referral hospitals, namely, the Pulmonary Department of Corfu General Hospital and the 5th Pulmonary Department of SOTIRIA Hospital for Chest Diseases. The study was conducted based on the Declaration of Helsinki on human subject studies and approved by local Institutional Review Boards (25—1 December 2021, consent form not required). Inclusion criteria were a COVID-19 diagnosis by polymerase chain reaction, WHO clinical progression scale ≥5, and treatment with tocilizumab according to national guidelines (severe COVID-19 pneumonia as defined above) [11]. All subjects were instructed to return for a follow-up visit at three months postdiagnosis (Day 0), and if that was not feasible, a telephone communication took place to establish 90-day mortality. The only exclusion criterion was the unavailability of data on survival at 3 months postdiagnosis. The majority of patients had a computed tomography (CT) scan of the chest, but that was not a prerequisite for inclusion in the study. Clinical predictors that were identified for three clinical outcomes (mortality, intubation, noninvasive ventilation (NIV)) were further explored for confirmation of findings in a separate validation cohort.

### 2.2. Clinical Investigations

Demographics, body mass index (BMI), COVID-19 vaccination status, comorbidities, and Charlson comorbidity index (CCI) were recorded. The day of tocilizumab treatment in relation to in-hospital days and disease onset and dose as well as concomitant treatments were also obtained. The partial arterial pressure of the oxygen/oxygen delivery concentration ratio (PO_2_/FiO_2_) was recorded on admission and on deterioration/clinical worsening day (designated as the day tocilizumab was administered, D_T_). The same applied for c-reactive protein (CRP), serum lactate dehydrogenase (LDH), d-dimers, ferritin, blood neutrophils, blood lymphocytes and neutrophil to lymphocyte ratio (NLR), and platelets. Aspartate aminotransferase (SGOT), alanine aminotransferase (SGPT), and troponin were recorded on admission alone, and procalcitonin (PCT) was recorded on D_T_. Charts were assessed for implementation of NIV (high flow nasal cannulae or noninvasive positive pressure ventilation), intubation and mechanical ventilation and all-cause mortality at 90 days. Discharge CRP was also recorded.

### 2.3. Radiology Investigations

In subjects for whom a CT scan was available, two dedicated pulmonologists (one from each department) scored the findings per quality (ground glass opacities, consolidation or mixed pattern), per quantity (occupying more or less than 50% of total lung fields) and per distribution (central perihilar, peripheral, or mixed).

### 2.4. Statistical Analysis

Continuous variables are presented as medians with 25%–75% interquartile ranges and were analyzed with an unpaired nonparametric two-tailed Mann–Whitney test. Categorical variables are presented as absolute numbers (frequency %) and were analyzed with Fischer’s exact test or the chi-square test. CCI is recorded numerically. Statistically significant variables were further analyzed with Cox proportional hazards multivariable regression analysis to identify predictors of event (s) (event = death, intubation, NIV). The log-likelihood ratio was examined for goodness of fit. Significant variables were used to plot survival curves using the log-rank (Mantel-Cox) test. Subjects were censored at 90 days. *p* and alpha values below 0.05 were considered significant. GraphPad Prism version 9.3.1 was used (San Diego, USA).

### 2.5. Validation Cohort

Predictors of event (s) recognized with Cox regression analysis were subsequently validated in a separate validation cohort consisting of retrospectively identified subjects hospitalized between September 2020 and December 2020 who were not treated with tocilizumab and were matched for sex and disease severity with study subjects. Predictors identified in the initial tocilizumab cohort were analyzed with ROC analysis to validate their performance per the same events (mortality, intubation, NIV).

## 3. Results

In total, 126 subjects fulfilling the inclusion criteria were initially recognized. Data for five subjects regarding 90-day mortality were not available and were excluded from the study. From the remaining 121 subjects analyzed, 96 were alive after 3 months (survivors), and 25 were deceased (non-survivors). The mortality rate was 20.6% (25/121). Demographics and baseline characteristics are shown in Table 1. Nonsurvivors were older, had more comorbidities, and were more likely intubated and on invasive ventilation. The timing and dose of tocilizumab did not vary between survivors and nonsurvivors. Most subjects received one dose of 8 mg/kg tocilizumab intravenously. The prophylactic dose of low-molecular-weight heparin showed a nonsignificant association with survival.

Clinical and radiological features are shown in Table 2. At the evaluation during hospital admission, survivors exhibited a higher P/F ratio and lower blood ferritin and troponin levels. Upon deterioration/clinical worsening (day of tocilizumab treatment, D_T_), survivors exhibited a higher P/F ratio, lower blood ferritin, higher blood lymphocytes, a lower N/L ratio, lower LDH, and less central disease distribution. Survivors also exhibited lower late post-tocilizumab CRP values.

All variables shown significant in univariate analysis with no more than 5 missing values were included in Cox proportional hazards multivariable regression analysis. The outcome was death from all causes, and cases were censored at 90 days. A model of 9 variables with no covariates was preferable (log-likelihood ratio (G squared): 63.05), with significant predictors of mortality being age and serum LDH on deterioration (D_T_), with CCI and admission ferritin marginally not significant (alpha < 0.05) (Table 3). Kaplan–Meier survival analysis using cut-offs for age and LDH combined showed worse survival in these subjects (log-rank test for trend *p* < 0.0001, Figure 1).

The same analysis with intubation as the outcome identified significant variables, including the P/F ratio on admission and LDH on D_T_, with a marginally nonsignificant central radiology distribution (Table 4). Last, regarding noninvasive ventilation as an outcome, significant variables were found to be the P/F ratio on admission and on D_T,_ LDH on D_T_, and central disease distribution (Table 5).

### Validation Cohort

This was a precedent without tocilizumab treatment retrospective cohort from our department of 50 COVID-19 cases matched for sex and severity (WHO CPS ≥ 5) to tocilizumab-treated study subjects. This cohort consisted of 32 males (32/50, 64%), who were significantly older than the study subjects (median age 74 years versus 63 years of age, *p* < 0.05) and had worse outcomes than the total tocilizumab group (mortality rate 19/50, 38% versus 25/121, 20.6%; intubation rate 16/50, 32% versus 26/121, 21%; NIV rate 27/50, 54% versus 27/121, 22%).

Predictors of mortality recognized in the tocilizumab group (Table 3, Table 4 and Table 5) significantly predicted adverse outcomes in the validation cohort by ROC analysis (Figure 2). An age above 77 years predicted death with 74% sensitivity and 81% specificity (AUC: 0.79). LDH upon clinical deterioration above 468 IU/L predicted death with 79% sensitivity and 77% specificity (AUC: 0.87). Furthermore, LDH on deterioration and P/F at admission both predicted intubation (AUC: 0.83, AUC: 0.80, respectively); P/F ratios at admission, at deterioration, and LDH at D_T_ all predicted NIV (AUC: 0.74, AUC: 0.88, AUC: 0.82, respectively) (all, *p* < 0.05).

## 4. Discussion

The main findings of this study are that among tocilizumab-treated patients with severe progressive COVID-19 pneumonia, older age, and particularly abnormal serum LDH on disease progression are associated with worse survival. These findings were confirmed in a separate cohort of critically ill patients irrespective of treatment with tocilizumab. Factors associated with intubation were P/F on admission and LDH upon disease progression, while the need for noninvasive ventilation was associated with P/F ratios on admission and on progression, serum LDH on progression, and central disease distribution on computed tomography scan.

First, the mortality rate observed in our study is similar to the rates reported in randomized controlled trials of tocilizumab but also in meta-analyses reporting a pooled mortality prevalence of approximately 20%–30% [5,6,7,9,12]. Previous studies have shown that age is a significant risk factor for COVID-19 mortality [14,15]. Age in general is shown to be the dominant risk factor for infections and associated mortality, through age-related hematopoietic mosaic chromosomal alterations leading to impaired immunity [16]. Other studies have identified inflammatory markers, such as the neutrophil to lymphocyte ratio, d-dimers, ferritin, and c-reactive protein, as markers of adverse outcomes [17]. However, data are lacking as to which patient responds to tocilizumab or not.

The discrepancies observed in positive trials (Recovery, REMAP-CAP) and negative trials (CORIMUNO-19, RCT-TCZ-COVID-19 Study Group, COV-AID study group) may be at least partially explained by our study [12,18,19]. CRP in our study was elevated, reflecting disease severity and the appropriate national administration guidance, but it was not associated with response to tocilizumab treatment, even when measured serially during the disease course. Similarly, in most negative studies, increased serum proinflammatory markers IL-6 and CRP did not guarantee a response to immunomodulatory treatment [20]. Taken together, these findings point to the necessary discovery of additional biomarkers to guide treatment.

In contrast, serum LDH has been reported by other authors at the beginning of the course of COVID-19 or to estimate disease resolution and not as a predictor of treatment responsiveness, as in our study [21]. Increased serum LDH ≥ 2× to 3× the upper limit of normal at the time of clinical worsening efficiently predicted all adverse outcomes in our study and was further validated in a severely compromised cohort. The TCZ dose and timing of administration (2nd in-hospital day, 10th day of symptoms) were identical between responders and nonresponders in our study and similar to previous literature. It is intriguing to suggest, based on these results, that older patients with rising LDH values, irrespective of CRP levels, might benefit from earlier treatment intensification with tocilizumab.

Lactate dehydrogenase (LDH) is a stable cytoplasmic enzyme that is found in all cells. LDH is rapidly released when the plasma membrane is damaged, a key feature of cells undergoing apoptosis, necrosis, and other forms of cellular damage [22]. Serum LDH is elevated due to wide spectrum of diseases including infections, cardiac, neurological, endocrinal, gastrointestinal, rheumatological, hematological, pulmonary diseases, and malignancies. LDH has already shown prognostic significance for COVID-19 [23]. LDH elevation in severe COVID-19 characterized by cytokine release syndrome reflects extended inflammasome activation of the Nod-Like Receptor Protein (NLRP-3) pathway, with other immune abnormalities including chemokines, Il-1β, nuclear factor kappa beta, IL-18, caspase-1, and damage associated molecular patterns (DAMPs). Thus, immunological abnormalities observed in severe COVID-19 may not be manipulated nor entirely effectively by tocilizumab [24,25,26]. This evidence may provide an explanation for the ineffectiveness of TCZ in our patients with high LDH, meaning with uncontrolled immune dysregulation and inflammation.

The P/F ratio is a crucial indicator for acute lung injury (≤300), acute respiratory distress syndrome (≤200), and severe ARDS (≤100). Responders in our study had a median P/F ratio of 150 when they received tocilizumab, while nonresponders had a significantly lower ratio of 100. Median P/F ratios of 280 on admission protected from subsequent intubation, while median P/F ratio of 150 upon clinical deterioration protected from the subsequent use of NIV. Negative studies employed patients with a ratio >200, while other investigators focused on patients with a P/F ratio <200 [13,18,19]. Studies that would evaluate “earlier” tocilizumab treatment based on a P/F ratio in the range of >100 to <200 would be useful, in our opinion. On the contrary, younger patients with serum LDH below a threshold (468 U/L in our study) and favorable P/F ratio >150 might not need rescue therapy with tocilizumab.

To our knowledge, the central disease distribution association with adverse outcomes is a novel finding not previously reported. Subpleural disease distribution has been linked with COVID-19 resolution [21]. This finding is in line with everyday clinical practice as well. This was significant for NIV but also, albeit not statistically, for the other two outcomes mortality and intubation. These patients might also need prioritization to treatment and could be the target of future clinical trials.

We acknowledge that our study has certain limitations, one being its retrospective design, lack of data posing restrictions in multivariable analysis, and absence of a control group. Furthermore, the adverse outcomes reported in this study (all-cause mortality) might be related not solely to severe COVID-19 but also to other etiologies (i.e., pulmonary embolism, sepsis, or concomitant other diseases). Last, radiology findings were classified by pulmonologists, since this was not designed as a radiology study.

On the other hand, our findings on LDH, P/F ratios and radiology are relatively novel and of clinical significance. Advantages of our study are the relatively large size of tocilizumab-treated subjects, the multicenter design, the 90-day data, the serially recorded clinical parameters, and the confirmation of our findings in a separate validation cohort. The high predictive value of our findings, as shown by the significant AUC values in the tocilizumab untreated validation cohort patients, implies that our findings relate to the disease (COVID-19) itself and may be tocilizumab independent.

## 5. Conclusions

In our retrospective multicenter study, death occurred in 20.6% of tocilizumab-treated severe COVID-19 patients. Predictors of mortality were increased age and particularly increasing serum levels of LDH during the disease course. Additional predictors of adverse outcomes were P/F ratios and central disease distribution. These findings should be used to define patient subpopulations in prospective clinical trials examining the efficacy of early therapy with tocilizumab or other agents.

## Figures and Tables

**Figure 1 vaccines-10-00978-f001:**
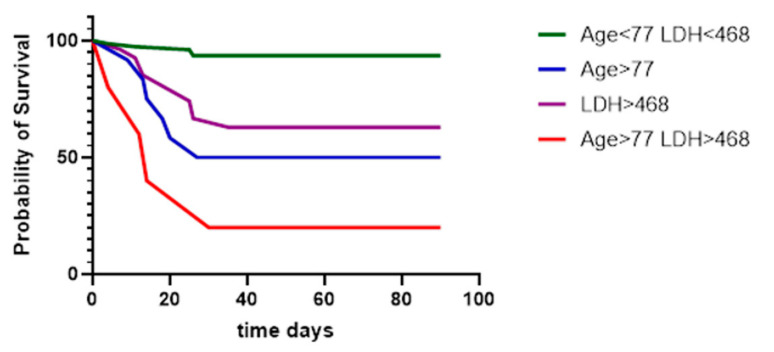
Kaplan–Meier survival curves of tocilizumab-treated subjects. Significantly worse survival was observed in subjects aged >77 years and with serum LDH on D_T_ >468 U/L (*p* < 0.0001).

**Figure 2 vaccines-10-00978-f002:**
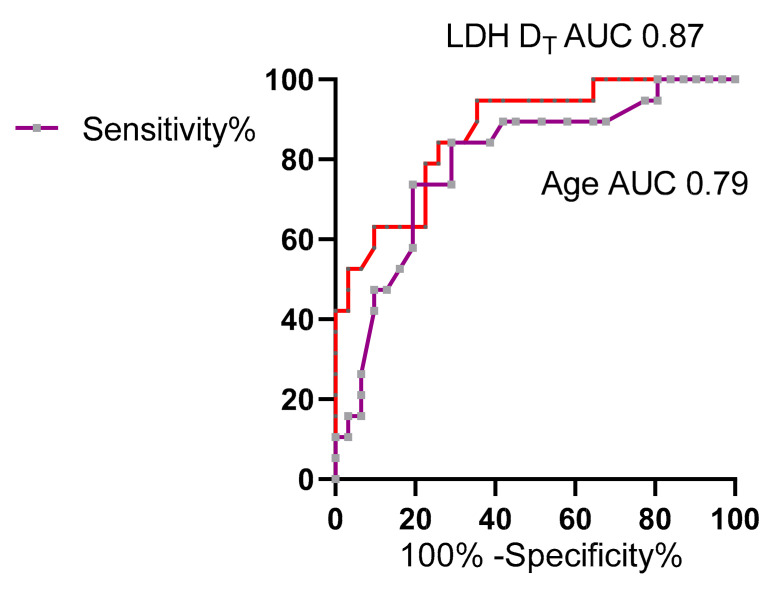
Predictors of mortality by ROC analysis of the validation cohort. LDH D_T_ had the largest AUC (red line) compared with the other predictors. Age, purple line (*p* < 0.05).

**Table 1 vaccines-10-00978-t001:** Study subjects’ demographics and treatment variables. Continuous variables are presented as medians (25th–75th IQR), and categorical variables are presented as n (frequency %). CCI: Charlson comorbidity index.

	Survivors (n = 96)	Non-Survivors (n = 25)	*p*
Age, years	57 (48–67)	75 (66–83)	<0.001
Male sex	60 (62.5)	15 (60)	0.9
Fully vaccinated	7 (7)	4 (16)	0.3
CCI	2 (1–3.7)	5 (3.5–7)	<0.001
Diseases:			
Chronic pulmonary disease	10 (10)	10 (40)	0.9
Cardiovascular disease	21 (21)	14 (56)	
Diabetes	14 (14)	10 (40)	
Chronic kidney disease	3 (3)	2 (8)	
Malignancies	6 (6)	2 (8)	
Connective tissue disease	3 (3)	2 (8)	
Body mass index	27 (25–30)	28 (26–29)	0.4
Hospitalization days	16 (12–21)	18 (11–26)	0.6
Hospital days until tocilizumab treatment	2 (1–4)	2 (2–3.5)	0.9
Symptom days until tocilizumab treatment	10 (8–12)	10 (6–12)	0.3
Tocilizumab total dose mg	600 (505–720)	640 (560–960)	0.17
Other treatments			
Remdesivir	54 (56)	17 (68)	0.3
Corticosteroids	96 (100)	25 (100)	0.9
Antibiotics	35 (36)	13 (52)	0.17
Anti-coagulation			
prophylactic dose	28 (29)	3 (12)	0.078
intermediate dose	53 (55)	14 (56)	
full therapeutic dose	15 (16)	8 (32)	
Non-invasive ventilation	20 (20)	7 (28)	0.4
Intubation-invasive ventilation	8 (8)	18 (72)	<0.001

**Table 2 vaccines-10-00978-t002:** Clinical and radiological features of study subjects. Continuous variables are presented as medians (25th–75th IQR), and categorical variables are presented as n (frequency %). P/F ratio: partial oxygen pressure/oxygen concentration; CRP: c-reactive protein; CPK: creatine kinase; LDH: lactate dehydrogenase; N/L ratio: blood neutrophils/blood lymphocytes; SGOT: aspartate aminotransferase; SGPT: alanine aminotransferase; ALP: alkaline phosphatase; γGT: glutathione-transferase; PCT: procalcitonin; TLF: total lung fields; GGO: ground glass opacities. In the case of missing values, n is depicted.

	Survivors (n = 96)	Non-Survivors (n = 25)	*p*
P/F admission	281 (238–314)	214 (188–271)	<0.001
Neutrophils/mm^3^ admission	4680 (3388–7410)	5740 (2990–10,725)	0.3
Lymphocytes/mm^3^ admission	895 (680–1175)	710 (495–1165)	0.12
N/L ratio admission	5.5 (3.8–8.2)	6.6 (3.4–16.5)	0.2
D-dimers ng/L admission	0.72 (0.51–1.1)	0.9 (0.47–1.54)	0.3
Ferritin admission (ng/mL)	587 (256–986) (n = 90)	1271 (480–2431)	0.004
Platelets × 10^3^/μL admission	180 (133–228)	175 (115–275)	0.8
CRP admission mg/L	8.7 (5–14)	8.3 (5.8–12.5)	0.8
CPK admission U/L	126 (79–219) (n = 50)	105 (69–246) (n = 20)	0.5
LDH admission U/L	342 (274–436)	413 (312–540)	0.09
SGOT admission U/L	36 (29–54)	38 (28–59)	0.8
SGPT admission U/L	35 (25–57)	31 (20–36)	0.076
ALP admission U/L	57 (50–78) (n = 50)	66 (57–78) (n = 20)	0.2
γ-GT admission U/L	42 (24–59) (n = 50)	28 (19–43) (n = 20)	0.078
Troponin ng/L admission	9.4 (5–15) (n = 50)	24 (11–30) (n = 20)	<0.001
P/F D_T_	150 (111–189)	100 (73–129)	<0.001
Neutrophils/mm^3^ D_T_	6880 (4100–9100)	8100 (5530–12,265)	0.10
Lymphocytes/mm^3^ D_T_	860 (605–1180)	560 (460–915)	0.0068
N/L ratio D_T_	6.8 (4.5–11.3)	16.6 (5–23)	0.0032
D-dimers ng/L D_T_	0.9 (0.6–1.6) (n = 60)	1.3 (0.7–2.7)	0.079
Ferritin D_T_ (ng/mL)	674 (457–1246) (n = 70)	1565 (529–2594)	0.022
Platelets × 10^3^/μL D_T_	223 (171–298)	222 (119–263)	0.2
CRP D_T_ mg/L	8.6 (5–13)	10.8 (4–14)	0.4
CPK D_T_ U/L	91 (48–147) (n = 50)	116 (77–213) (n = 20)	0.19
LDH D_T_ U/L	354 (297–444)	530 (369–665)	<0.001
PCT ng/mL D_T_	0.1 (0.02–1.1) (n = 10)	0.5 (0.07–1.2) (n = 5)	0.6
CRP late (time of discharge or death) mg/L	0.09 (0.0–0.4) (n = 94)	7.6 (2.6–27) (n = 22)	<0.001
Radiology	n = 92	n = 24	
Extent > 50% TLF	65 (70)	21 (87)	0.15
Central/peripheral/mixed	5/27/60	5/2/17	0.013
Ggo/consolidation/mixed	33/8/51	8/2/14	0.9

**Table 3 vaccines-10-00978-t003:** Cox regression multivariable analysis (outcome: death). P/F ratio: partial arterial oxygen pressure/oxygen concentration; LDH: lactate dehydrogenase; D_T_: day of tocilizumab treatment.

Variable	Hazard Ratio	95% Confidence Interval	*p*
Age	1.082	1.01 to 1.15	0.010
Charlson comorbidity index	1.30	0.97 to 1.71	0.063
P/F admission	1.000	0.99 to 1.009	0.95
Ferritin admission	1.000	1.000 to 1.000	0.0504
Lymphocytes D_T_	0.999	0.999 to 1.000	0.34
LDH D_T_	1.004	1.002 to 1.007	0.0013
P/F D_T_	0.991	0.97 to 1.002	0.29
Mixed distribution	1.744	0.382 to 18.02	0.54
Central distribution	3.769	0.538 to 43.22	0.211

**Table 4 vaccines-10-00978-t004:** Cox regression multivariable analysis (outcome: intubation) (log-likelihood ratio: 37). P/F ratio: partial arterial oxygen pressure/oxygen concentration; N/L: blood neutrophils/blood lymphocytes; LDH: lactate dehydrogenase; D_T_: day of tocilizumab treatment.

Variable	Hazard Ratio	95% Confidence Interval	*p*
Age	1.005	0.95 to 1.06	0.85
Charlson comorbidity index	1.018	0.76 to 1.31	0.89
P/F admission	0.990	0.985 to 0.996	0.001
Ferritin admission	1.000	0.999 to 1.000	0.12
N/L D_T_	1.033	0.99 to 1.07	0.11
LDH D_T_	1.003	1.000 to 1.005	0.019
P/F D_T_	1.00	0.995 to 1.005	0.99
Mixed distribution	1.66	0.47 to 8.99	0.48
Central distribution	5.16	0.95 to 33.05	0.058

**Table 5 vaccines-10-00978-t005:** Cox regression multivariable analysis (outcome: noninvasive ventilation) (log-likelihood ratio: 44). P/F ratio: partial arterial oxygen pressure/oxygen concentration; N/L: blood neutrophils/blood lymphocytes; LDH: lactate dehydrogenase; D_T_: day of tocilizumab treatment.

Variable	Hazard Ratio	95% Confidence Interval	*p*
Age	1.005	0.96 to 1.04	0.78
Charlson comorbidity index	0.913	0.726 to 1.126	0.41
P/F admission	0.993	0.988 to 0.998	0.005
Ferritin admission	1.000	0.999 to 1.000	0.86
N/L D_T_	1.029	0.99 to 1.06	0.11
LDH D_T_	1.002	1.00 to 1.004	0.017
P/F D_T_	0.992	0.985 to 0.998	0.036
Mixed distribution	2.066	0.84 to 6.31	0.15
Central distribution	4.49	1.27 to 16.96	0.020

## Data Availability

The data presented in this study are available on request from the corresponding author. The data are not publicly available due to ethical reasons.

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
