# Peer review of "Predictors of Mortality in Tocilizumab-Treated Severe COVID-19"

_vaccines, 2022, doi:10.3390/vaccines10060978_

Round 1

Reviewer 1 Report

Major comments

This paper has attempted to clarify some of the factors related to mortality in patients with severe COVID-19 treated by tocilizumab (TCZ).

The primary mortality-related factors were shown to be older age and an elevated serum lactate dehydrogenase (LDH) concentration.

It is important to demonstrate that age >77 years and LDH concentration >468 are significantly associated with the prognosis regardless of TCZ administration.

The presence or absence of TCZ is not a major reason for publication of a paper. 

However, these factors have already appeared in other documents written by the other author.

Please see that suggestion.

I would like the authors to consider why the LDH concentration is associated with the prognosis.

I would also like the authors to consider why TCZ is ineffective for patients with an elevated LDH concentration.

The key point of this document is the following, as mentioned in the author’s discussion: “It is intriguing to suggest, based on these results, that older patients with rising LDH values, irrespective of CRP levels, might benefit from earlier treatment intensification with tocilizumab.”

If the above hypothesis can be proven, publication of this article would be worthwhile.

Alternatively, it is recommended to make a conclusion based mainly on earlier treatment intensification with tocilizumab.

Minor comments

1) What does it mean to increase LDH?

2) Please consider why TCZ is ineffective in patients with a high LDH concentration.

3) In Tables 1 and 2, the rows containing the element and the rows containing the resulting numerical values are out of alignment.

4) For age and prognosis, please refer to the following paper: “Hematopoietic mosaic chromosomal alterations increase the risk for diverse types of infection Nature Medicine DOI 10.1038 / s41591-021-01371-0.”

5) Line 234 reads, “P/F ratios of >200 on admission protected from intubation, P/F>150 upon clinical deterioration protected from the use of NIV.” Is that true?

6) Line 240 reads “subpleural radiology.” Is this the correct term?

Reviewer 2 Report

I considered the manuscript entitled “Predictors of mortality in tocilizumab-treated severe COVID-19” by Konstantinos Pagkratis, et al that is intended to be published in Vaccines journal.

Summary: Present manuscript deals with the retrospective evaluation of the fate of severe disease Covid19 patients who underwent Tocilizumab treatment as rescue therapy. It is usually difficult to analyze these patients as they are a heterogeneous population, but authors gathered them nicely. Results are convincing. It is obvious that non survivors are those that present worse inflammatory parameters, but the authors attest to this fact. This defines a small population that may not need to be treated. A decision tree should be constructed.

Strengths and weaknesses. One strength of the study is that authors present in the introduction the full scope of the Tocilizumab success in Covid19 patients for the moment and positions adequately the problem that they study. It is an unmet need for the Tocilizumab treatment in those patients. And agree with authors concerning: Advantages of our study are the relatively large size of tocilizumab-treated subjects, the multicenter design, the 90-day data, the serially recorded clinical parameters and the confirmation of our findings in a separate validation cohort

Major Concerns: none

Minor Concerns: none
